# Modulating the Activity of Androgen Receptor for Treating Breast Cancer

**DOI:** 10.3390/ijms232315342

**Published:** 2022-12-05

**Authors:** Chan-Ping You, Ho Tsoi, Ellen P. S. Man, Man-Hong Leung, Ui-Soon Khoo

**Affiliations:** Department of Pathology, Rm 014, 7/F, Block T, Queen Mary Hospital, Pokfulam Road, School of Clinical Medicine, Li Ka Shing Faculty of Medicine, The University of Hong Kong, Hong Kong SAR, China

**Keywords:** breast cancer, androgen receptor, targeted therapy, novel treatment, prognosis

## Abstract

The androgen receptor (AR) is a steroid hormone receptor widely detected in breast cancer. Evidence suggests that the AR might be a tumor suppressor in estrogen receptor alpha-positive (ERα+ve) breast cancer but a tumor promoter in estrogen receptor alpha-negative (ERα-ve) breast cancer. Modulating AR activity could be a potential strategy for treating breast cancer. For ERα+ve breast cancer, activation of the AR had been demonstrated to suppress the disease. In contrast, for ERα-ve breast cancer, blocking the AR could confer better prognosis to patients. These studies support the feasibility of utilizing AR modulators as anti-cancer drugs for different subtypes of breast cancer patients. Nevertheless, several issues still need to be addressed, such as the lack of standardization in the determination of AR positivity and the presence of AR splice variants. In future, the inclusion of the AR status in the breast cancer report at the time of diagnosis might help improve disease classification and treatment decision, thereby providing additional treatment strategies for breast cancer.

## 1. Introduction

The androgen receptor (AR), also known as NR3C4, is one of the sex steroid hormone receptors widely expressed in males and females. The receptor plays a vital role in multiple physiological processes, such as the development of the reproductive system [1]. The *AR* gene is located at locus Xq11–12 on the X-chromosome. It harbors eight exons encoding a protein of 110 kDa molecular weight. The AR protein is composed of three main functional domains: the N-terminal domain (NTD) (encoded by exon 1), the DNA binding domain (DBD) (encoded by exon 2–3) and the C-terminal ligand binding domain (LBD) (encoded by exon 5–8), together with a hinge region (HR) (encoded by exon 4) [2]. The inactivated AR is predominantly localized to the cytoplasm where it binds to molecular chaperones and co-chaperones which stabilize its protein structure and maintain it in an inactivated state [3]. The AR binds to ligands, such as dihydrotestosterone (DHT), via its LBD. This binding induces a conformational change and dissociation with its chaperones, leading to AR activation. As a transcriptional factor, the active AR is rapidly translocated into the nucleus and utilizes its DBD to recognize and bind to androgen-receptor-responsive elements (AREs) in the promoter or enhancer of its target genes. Such an interaction can modulate the transcriptional activity of the target genes [4]. A cytoplasmic AR can also interact with and trigger several secondary message pathways, leading to non-genomic activity. The AR targets a diverse range of oncogenic pathways, including PI3K/AKT, EGFR, Src and WNT pathways [5,6,7,8]. These pathways cover various cellular processes such as cell proliferation, migration, metastasis, apoptosis, differentiation and DNA damage repair. Steroid hormone receptors, such as estrogen receptor alpha (ERα), are considered a critical factor in the development of breast cancer [9]. Likewise, dysregulation of the AR also has been correlated to carcinogenesis. It is noted that around 80–90% of prostate cancers are detected as AR-positive (AR+ve) at the time of diagnosis [10], and AR signaling is crucial for prostate cancer development and progression [11]. Androgen deprivation therapy by drugs such as AR antagonists or by surgical castration to block the activity of the AR has become a gold standard in controlling disease progression. Furthermore, the AR may contribute to the growth of other malignancies, such as breast cancer, hepatocellular carcinoma and ovarian cancer, with ongoing clinical trials targeting the AR in patients bearing these cancers [12]. Therein, this narrative review will discuss the potential for targeting the AR in breast cancer treatment. The suitable electronic sources used in the current review were identified from the PubMed database by searching the following key words: breast cancer; androgen receptor; targeted therapy; prognosis.

## 2. Different Roles of AR in Breast Cancer

According to GLOBOCAN 2020, the latest global cancer statistics reported by the International Agency for Research on Cancer (IARC) of the World Health Organization (WHO), female breast cancer was estimated to be the most prevalent cancer worldwide, with approximately 2.26 million newly diagnosed cases (accounting for 11.7% of all cancers) and 0.68 million death cases in 2020 [13]. Breast cancer is a highly heterogeneous tumor that can be further categorized into five subtypes depending on their intrinsic gene expression profiles: Luminal A, Luminal B, human epidermal growth factor receptor 2-enriched (HER2+ve), basal-like and normal-like [14], of which Luminal A, Luminal B and normal-like are ERα-positive (ERα+ve), HER2+ve are ERα negative (ERα-ve) and basal-like are often ERα, progesterone receptor (PR) and HER2 negative (i.e., triple negative breast cancer; hereafter referred as TNBC). Remarkably, it has been reported that around 70–90% of breast cancers are detected as AR+ve cases [15,16], implicating that the AR may play some roles in disease development and progression. However, the role of the AR in breast cancer is controversial.

In ERα+ve breast cancer, ERα-mediated signaling is the driving factor for cancer development. The AR, by interfering with the function of ERα, may have a tumor suppressor function. The importance of cross-talk between AR and ERα has been supported by mechanistic studies which suggest that AR and ERα can interact with each other through the AR’s N-terminal domain (NTD) and ERα’s ligand-binding domain (LBD). This protein–protein interaction can inhibit the activity of both proteins [17]. Studies have also reported that the AR can compete with ERα by binding to Erα-targeted DNA sequences (Erα-responsive elements, ERE) or by interacting with its co-activators to hijack ERα from ERE to ARE (Figure 1) [18,19]. The AR can also up-regulate tumor suppressor genes, such as the tumor suppressors *SEC14L2*, *EAF2* and *ZBTB16*, to repress tumor growth directly [19]. Furthermore, the AR can up-regulate the expression of the tumor suppressor protein, estrogen receptors beta (ERβ), to indirectly inhibit the biological activity of ERα and down-regulate the expression level of the proto-oncogene *CCND1*, or may enhance other tumor suppressors, including *FOXO1* and *FOXO3a* [20,21]. Due to these activities against ERα, patients with an ERα+ve/AR+ve phenotype would have relatively better clinical features, e.g., smaller tumor [22,23], and longer survival time indicated by disease-free survival (DFS), overall survival (OS) and recurrence-free survival (RFS) [24,25,26]. Thus, the AR should be a good prognostic factor in ERα+ve breast cancer.

In ERα-ve breast cancer patients, including HER2+ve and TNBC, the AR shows oncogenic effects. In HER2+ve breast cancer, it has been reported that an active AR could induce the expression of HER2, which subsequently activates the MAPK pathway. The activated MAPK pathway induces the AR’s expression, resulting in a positive-feedback loop [27]. Moreover, an activated AR could up-regulate *WNT7B*. Activation of the WNT/β-catenin pathway would result. In addition, the AR could further form a complex with β-catenin. The complex could promote the transcription of *HER3*, which is a critical factor for the oncogenic activity of HER2 [28]. In TNBC, the AR has been shown to interact with SRC directly and activate the oncogenic pathway SRC/PI3K/FAK and its downstream genes [29]. These actions contribute to the continuous development of the tumor, leading to poor survival of the patients, rendering the AR a poor prognostic factor in ERα-ve breast cancer [25,30,31,32,33].

## 3. Modulating the Activity of AR as Breast Cancer Treatment

### 3.1. Activation of AR Can Suppress ERα+ve Breast Cancer

The possibility of modulating AR activity for breast cancer treatment can be traced back to 1939, when the first record of breast cancer patients who might benefit from receiving testosterone propionate to activate the AR was published [34]. Subsequently, similar observations were reported to support the potential application of androgens in treating breast cancer [35,36,37,38]. Although the expression status of ERα in the tumors was not indicated in these studies, they indicated the therapeutic value of utilizing androgens in breast cancer patients. Nevertheless, this innovative therapeutic strategy for breast cancer has not been widely studied for decades, since people found that androgens could be converted to estrogen via steroid aromatase. Such an effect may enhance the activity of the oncoprotein ERα. As more and more evidence revealed the anti-ERα properties of AR in breast cancers, investigators started to revisit the potential of androgens in treating ERα breast patients.

In a retrospective study, 508 postmenopausal and ovariectomized women who had received testosterone implants in addition to conventional hormone therapy were followed up for a mean duration of 5.8 years. The incidence of breast cancer was recorded, and the results showed that using testosterone might reduce the occurrence of conventional-hormone-therapy-induced breast cancer [39]. A prospective study recruiting 1268 pre-/postmenopausal women who were subcutaneously implanted with testosterone alone or combined with an aromatase inhibitor anastrozole to treat symptoms of hormone deficiency aimed to study the influence of the treatments on the occurrence of breast cancer: the 5-year interim report indicated that the annual incidence rate in the intent to treat patients was estimated to be 142 cases per 100,000 individuals (0.142%) and it could be further reduced to 73 cases per 100,000 people (0.073%) if the subjects received testosterone therapy, which was significantly lower than the control groups (0.293–0.39%) [40]; the 10-year analysis results showed the incidence rate of breast cancer in the testosterone-treated population was also significantly less than the age-specific Surveillance, Epidemiology, and End Results (SEER) expected result (0.165% vs. 0.271%). This study supported the role of testosterone in lowering the risk of hormone-therapy-related breast cancer [41].

The combination of testosterone and anastrozole was also applied to treat menopausal symptoms in 72 breast cancer survivors. The treatment was effective in relieving the symptoms without elevating estradiol; no tumor relapse was found for up to 8 years [42]. Combining testosterone and anastrozole for treating a hormone-receptor-positive breast cancer patient presented a promising result with a 12-fold decrease in the tumor volume and without elevation of estradiol [43]. Similarly, the treatment of testosterone and another aromatase inhibitor, letrozole, in hormone-receptor-positive invasive breast cancer patients led to 43% diminished tumor size with complete pathologic response when given in combination with chemotherapy [44].

Another study provided more direct evidence to support that androgens as a single drug could be an effective anti-cancer agent for treating ERα+ve breast cancer patients. The study recruited 53 ERα+ve/PR+ve metastatic breast cancer patients whose tumors no longer responded to anti-hormonal therapies. These patients were treated with testosterone propionate. Approximately 60% of the subjects showed disease regression or stabilization, representing a positive result [45]. An independent group performed a similar study. Hormone-receptor-positive metastatic breast cancer patients were subjected to receiving another androgen, fluoxymesterone. The results showed that around 43% of the participants achieved clinical benefit (3% complete response, 10% partial response and 30% stable disease) for at least 6 months [46].

As proof of principle, a phase II trial (NCT01616758) on evaluating the efficacy and safety of enobosarm, a synthetic and selective AR activating agent, in ERα+ve metastatic breast cancer demonstrated that around 50% (8/16) of the patients reached the best response with a median duration of 4.5 months. The drug was well tolerated [47]. Due to the promising result, a larger-scale trial (NCT02463032) was subsequently conducted. In the enobosarm-treated ERα+ve metastatic breast cancer patients whose nuclei AR staining is more than 40%, the clinical benefit rate (CBR), the best objective tumor response (BOR) and the median radiographic progression-free survival (rPFS) were 80%, 48% and 5.47 months (mean = 7.15 months), respectively. In comparison, in the patients whose nuclei AR staining is less than 40%, the CBR, BOR and rPFS were 18%, 0% and 2.72 months (mean = 2.7 months), respectively, suggesting the anti-cancer potential of enobosarm in ERα+ve breast cancer [48]. To further confirm these results in a larger cohort, a phase III study (NCT04869943) to evaluate the efficacy of enobosarm in AR+ve/ERα+ve metastatic breast cancer patients has been recently launched and is ongoing.

More recently, a pre-clinical study also demonstrated that activation of the AR by either DHT or enobosarm in ERα+ve endocrine-resistant breast cancer patient-derived xenografts (PDXs) significantly suppressed the estrogen-induced tumor growth. By contrast, inhibition of the AR activity via the AR antagonist enzalutamide enhanced the ability of estrogen to stimulate tumor growth [19]. These studies reflected the safety and efficacy of androgens in decreasing the risk of hormone-induced breast cancer, suppressing the tumor progression and inhibiting tumor growth, indicating the feasibility of utilizing androgens for treating ERα+ve breast cancer patients.

### 3.2. Blocking of AR Can Suppress ERα-ve Breast Cancer

#### 3.2.1. Evidence from Clinical Studies

Many AR suppressive agents are available for treating prostate cancer. These agents significantly contributed to improving patients’ treatment outcomes. Due to the importance of the AR in ERα-ve breast cancer, studies on evaluating the efficacy of anti-androgens, alone or combined with other anti-breast cancer drugs, have been documented. A phase II clinical trial (NCT02091960) was conducted to evaluate the therapeutic value of the AR antagonist enzalutamide. In the trial, enzalutamide was used on advanced ERα-ve/AR+ve/HER2+ve breast cancer patients. In all, 24% of the patients exhibited a clinical benefit at 24 weeks [49]. A case report described a 55-year-old patient with metastatic TNBC with 100% of nuclei showing a positive signal for AR staining revealed by IHC. The patient received seven lines of cytotoxic chemotherapy and radiotherapy for which only a partial response could be reached, and the disease progressed. The anti-androgen drug bicalutamide was then introduced, to which the patient achieved a complete clinical response at four months and sustained for eight more months [50].

Another phase II study (NCT00468715) demonstrated bicalutamide in AR+ve/ERα-ve/PgR-ve metastatic breast cancer. The results showed a clinical benefit rate of 19% at 6 months and a median progression-free survival of 12 weeks, comparable to other chemotherapies studies in TNBC patients [51]. Similarly, a study (NCT01889238) evaluated the effect of enzalutamide on locally advanced or metastatic AR+ve/TNBC. A clinical benefit rate of 33% was achieved with improved progression-free survival (3.3 months) and overall survival (17.6 months) [52].

An ongoing phase II trial (NCT03383679) combined an androgen-receptor antagonist darolutamide with a chemo-drug capecitabine in advanced AR+ve/TNBC patients recently reported the first-stage result of the study that darolutamide is well tolerated and 26.3% of the patients present a clinical benefit at 16 weeks. The project is now moving to its second stage [53]. Another ongoing phase IIB trial project (NCT02689427) applying enzalutamide in combination with the chemo-drug paclitaxel to treat AR+ve/TNBC patients reported that 33.3% of the patients who were resistant to the doxorubicin-based drugs showed complete response or minimal residual disease to the treatment. This value was lowered to 23% in patients with luminal androgen receptor (LAR). The results suggested a unique pathological feature in this sub-population; noteworthy, around 85% of the LAR TNBC patients were detected to have aberrant PI3K pathway activation in the study. Hence, additional PI3K-targeted drugs for patients who belong to this specific subtype may significantly enhance the treatment outcomes [54].

Other than AR inhibitory drugs, agents targeting androgen synthesis also showed promising potential in clinical application for TNBC patient management. Abiraterone acetate is a CYP17A1-specific inhibitor that suppresses the production of androgens [55]. Combined abiraterone acetate and prednisone for treating locally advanced or metastatic AR+ve/TNBC patients in a phase II trial (NCT01842321) have been tested. The results demonstrated that the clinical benefit rate at 6 months was 20.0%, with one subject having a complete response and five subjects with stable disease for more than 6 months. The median progression-free survival was 2.8 months [56].

A novel antiandrogenic drug seviteronel which functions by a unique dual mechanism of action that lowers the biosynthesis of androgen via inhibiting CYP17 lyase, as well as blocking AR activation via antagonizing AR, has attracted great attention since the drug is more effective than abiraterone acetate in suppressing AR activity [57]. An open-label phase I study proved that seviteronel was well-tolerated at the dose of 450 mg once daily. Of the enrolled TNBC patients, 28.6% (2/7) under this dose reached clinical benefit at 4 months, indicating the drug’s safety and efficacy in treating the patients [58]. In stage I of the phase II study (NCT02580448), the preliminary result showed that 33% of the TNBC subjects could achieve a clinical benefit at 4 months, which was similar to the study mentioned above; moreover, 70% of the subjects presented with decreased circulating tumor cells after the treatment. The results confirmed the clinical activity of seviteronel in breast cancer therapy [59]. Furthermore, multiple clinical trials (NCT03004534, NCT02457910, NCT03207529, NCT03090165, NCT05095207, NCT01990209, NCT04947189) are underway investigating the feasibility of AR-targeting therapy, alone or combined with other therapeutic agents, in AR+ve/ERα-ve breast cancer patients. These results highlight the feasibility of using anti-androgen agents for treating ER-ve breast cancer.

#### 3.2.2. Evidence from Pre-Clinical Studies

In pre-clinical studies, blocking AR signaling in ERα-ve breast cancer models has been shown to suppress cell growth. Inhibition of AR activity by shRNA or enzalutamide could interfere with cell proliferation in AR+ve/ERα-ve/HER2+ve breast cancer cell lines and xenograft models; when combined with trastuzumab, it could further enhance the suppressive capacity. Enzalutamide could reduce the expression of the proliferation marker Ki-67 and induce the expression of active caspase-3 in the xenograft model, demonstrating an anti-proliferative property after AR suppression in HER2+ve breast cancer [60].

A combination of enzalutamide and trastuzumab or mTOR inhibitor everolimus in AR+ve/ERα-ve/HER2+ve and AR+ve/TNBC could synergistically inhibit cell proliferation [61]. Additionally, inhibition of AR in a trastuzumab-resistant HER2+ve breast cancer cell line has been shown to suppress cell proliferation effectively and re-sensitize cells to trastuzumab [61]. It has been shown that concurrent treatment of TNBC cells with bicalutamide and EGFR, PDGFRβ and Erk1/2 inhibitors or PI3K inhibitor could additively inhibit cell proliferation; suppression of the PI3K/mTOR pathway in the cells was demonstrated to decrease the expression of the AR, which indicates a potential direction for anti-androgen drug development for treatment of TNBC patients [62,63].

Blocking the AR could significantly suppress proliferation and increase apoptosis in both in vitro and in vivo AR+ve/ERα-ve models, which might result from the inhibition of Wnt/β-catenin and EGFR signaling pathways [64,65]. ERα-ve cases tend to have more aggressive clinical features, such as higher tumor grade and metastatic nature. Unfortunately, the option for available targeted therapies for this population is limited. Patients who initially responded to treatment quickly relapse later. Therefore, the patients usually have a poor prognosis [66]. The introduction of anti-androgen drugs in these subtypes of patients may help to relieve this predicament.

## 4. Challenges in Targeting AR for Breast Cancer Treatment

### 4.1. Lacking Consensus on Defining the AR Positivity in Breast Cancer Hampers the Clinical Diagnosis of the Marker in the Patients

Unlike other prognostic and predictive factors such as ERα that have been standardized for detection in breast cancer, the AR detection protocol to define positivity in patients has not been standardized. The cut-off scores for pathological examination by immunohistochemistry (IHC) of AR positivity varied in different studies. Moreover, the antibodies against the AR used for IHC staining in different studies have been diverse (Table 1). Other studies determined AR positivity by assessing the mRNA of the AR [67,68]. These could very well be the reasons that has led to contradictory conclusions with resulting controversy amongst studies defining the role of the AR in breast cancer patients. Before introducing AR-targeting therapies for breast cancer treatment in the clinical setting, a standardized procedure for AR detection and an agreed definition for AR positivity needs to be established.

**Table 1 ijms-23-15342-t001:** The definition of AR positivity as determined by IHC, and the antibodies used varied in different studies.

Anti-AR Antibody	Definitions of AR Positivity	Company; Catalog Number	References
AR441	Nuclear stained ≥ 10%	Thermo Scientific, Fremont, CA, USA; MA5-13426	[25]
	Nuclear stained ≥ 1%	Dako, Glostrup, Denmark; M3562	[24,26,30,32,67,69]
	H-score > 10		[70]
	Nuclear stained ≥ 75%		[71]
	Nuclear stained ≥ 10%		[72]
F39.4.1	H-score ≥ 190	BioGenex, Fremont, CA, USA; AM256-5ME	[23]
	≥1% ^#^		[73]
AR27	Nuclear stained ≥ 10%	Leica, Newcastle, UK; AR-318-L-CE	[74]
SP107	Nuclear stained ≥ 1% and ≥10%	Cell Marque, Rocklin, CA, USA; 200R-14	[75]
EP120	Nuclear stained ≥ 10%	ZSGB, Beijing, China; ZA-0554	[76]
EPR1535 (2)	≥10% ^#^	Abcam, Cambridge, UK; ab133273	[77]
ER179 (2)	Nuclear stained ≥ 1%	Epitomics, Burlingame, CA, USA; 3184-1	[31]

^#^ Indicates that AR expression was not defined in either nuclear or cytoplasmic compartments in the original study.

### 4.2. The AR Splice Variants (AR-Vs) Might Affect the Outcomes of AR-Targeting Therapies

Alternative splicing of the AR can give rise to variants, some with truncation of the LBD. These variants may be innately and constitutively active even in the absence of ligands. Based on current studies, at least 18 AR-Vs have been identified (Figure 2), many of which have been correlated to castration resistance, tumor metastasis and subsequently poor OS and DFS in prostate patients, leading to failure of treatment [78,79]. For example, AR-V7, a constitutively active AR variant, has been implicated as the most clinically relevant splice variant in prostate cancer patients. The patients are usually resistant to anti-AR therapy and have poor prognoses [80,81]. Moreover, evidence suggests that targeting AR-V7 may effectively re-sensitize the tumor cell to AR-targeting treatments or inhibit tumor cell proliferation, supporting the potential application of targeting AR-Vs for improving treatment outcome in AR-dominant cancer patients [82,83,84,85,86]. Hence, to address this issue, agents for targeting AR-Vs may need to be considered for the development of AR-targeting strategies in the treatment of breast cancers.

Fortunately, several strategies for targeting AR-Vs have been established [87] for the development of anti-AR-V drugs for which clinical trials have been conducted. For instance, TAS3681, a selective AR antagonist that can reduce the protein expression levels of AR and AR-V7 [88], has been tested in metastatic castration-resistant prostate cancer (NCT02566772). The efficacy of niclosamide, an AR-V7 inhibitor [83], in combination with other anti-prostate cancer agents, has also been tested in advanced prostate cancer (NCT03123978) and hormone-resistant prostate cancer (NCT02807805). However, these agents are not yet available in the clinic.

**Figure 2 ijms-23-15342-f002:**
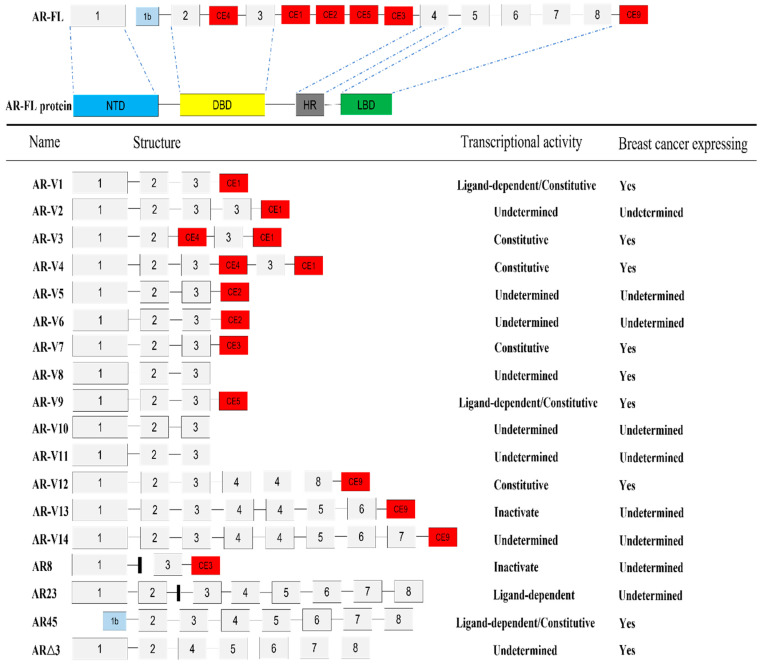
The wild-type AR (full-length) and its alternative splice variants. The full-length AR (AR-FL) gene harbors eight canonical exons, exons 1–8 and seven cryptic exons (CEs), 1b, CE1-5 and CE9. Different transcriptional manners of the gene generate AR-FL and at least eighteen AR-Vs, including AR-V1 [89], AR-V2 [90], AR-V3 [90], AR-V4 [89], AR-V5 [90], AR-V6 [90], AR-V7 [89], AR-V8 [91], AR-V9 [91], AR-V10 [91], AR-V11 [91], AR-V12 [92], AR-V13 [92], AR-V14 [92], AR8 [93], AR23 [94], AR45 [95] and ARΔ3 [96]. Among them, AR-V5 and AR-V6 have similar structures but differ by an 80 bp contiguous 5′ extension in the CE2 of AR-V5 [90]; AR-V8, AR-V10 and AR-V11 also have similar structures but differ by their downstream 3′ sequence with the proteins truncated after exon 3 with 10–39 amino acids extension before the stop codon [91]. AR8 and AR23 contain a unique sequence (presented by a small black square in the figure) with 23 amino acids inserted in the DBD [93,94].

### 4.3. High Expression Levels of AR in ERα+ve Patients May also Lead to Poor Prognosis in Disease Management

The AR tends to be an ERα suppressor in breast cancer. Activating the AR properly in AR+ve/ERα+ve patients to counteract the activity of ERα should be a prioritized option for controlling disease progression. However, without the influence of ERα, such as in ERα-ve breast cancers, the AR can also promote disease progression. It has been reported that breast tumors with an AR-to-ERα (AR/ERα) ratio greater than or equal to 2 have higher expression of proliferation signatures (*AURKA*, *BIRC5*, *CCNB1*, *MKI67* and *UBE2C*) than those with a value less than 2 [97]. Aggressive clinical features such as lymph node metastasis, larger tumor size or higher histopathological grade [98] and poorer response to tamoxifen have been associated with a high AR-to-ERα ratio [99].

Moreover, independent studies focused on different AR/ERα ratios also observed similar results. For instance, patients with a higher AR/ERα ratio value have a poorer prognosis, shorter disease-specific survival (DSS) and a higher risk of tumor aggression in breast cancer clinical studies [100,101]. Although activation of AR has been shown to suppress ERα+ve breast cancer by counteracting the activity of ERα, we should bear in mind that AR could have an oncogenic effect, especially in those patients with higher values of AR/ERα ratio. In such a condition, the tumor-driving ability of AR may become dominant. Thus, maintaining the balance between the signal intensity of AR and ERα pathways to reach an optimal therapeutic effect will need serious consideration for the further application of targeting the AR in breast cancer treatment.

### 4.4. The Current Molecular Subtyping of Breast Cancer May Not Reflect the Role of AR Accurately in Breast Cancer

The different roles of AR in ERα+ve and ERα-ve breast cancers have been discussed separately. As a heterogeneous cancer, the different intrinsic molecular profiles of breast cancer provide the basis for breast cancer molecular subtyping, which largely determines treatment strategy for each patient. Although the AR is expressed in a considerable proportion of breast cancer, due to its undetermined role, the expression status of AR has not been included in the current molecular subtyping. However, novel molecular subtypes of TNBC involving the AR have identified a subtype called the luminal androgen receptor (LAR) [102]. Interestingly, this subtype demonstrated the worst response to neoadjuvant chemotherapy but the best prognosis amongst the different TNBC subtypes [103], despite its AR being highly expressed (>10-fold). Moreover, as mentioned earlier, high expression of AR in ERα+ve breast cancer patients (high AR/ERα ratio) are associated with worse prognosis. These findings indicate that the current molecular subtyping system for breast cancer might not be sufficient to define the roles of the AR in breast cancer. Hence, to achieve an optimal therapeutic outcome by targeting the AR, a more detailed breast cancer molecular subtyping needs to be proposed, with the expression status of the AR in breast cancer taken into account.

### 4.5. Side Effects of Modulating AR Activity

Since the AR plays a critical role in multiple normal biological processes, it is not surprising that treating breast cancer patients with AR modulators, either agonist or antagonist, will alter the activity of the AR. This may have adverse effects. AR agonists have been reported to induce acne, hirsutism and alteration in lipid profile [104], while AR antagonists may cause adverse effects such as amenorrhea, hyperkalemia, gynecomastia and thromboembolism [105]. These side effects may decrease the patient’s quality of life or even be life threatening. Before using these agents for treating breast cancer, we must know if the patients will gain a benefit that can compensate for the side-effects.

## 5. Future Perspectives

Currently, many conventional AR-targeting drugs that modulate the activity of AR signaling can be applied in cancer therapy. Patients who benefit from these drugs have largely had their survival time prolonged, with improved disease treatment outcomes, regardless of their standard treatment for prostate cancer or their clinical trial treatment for breast cancer. In addition, more novel drugs produced by advanced techniques are in development. The proteolysis-targeting chimaera (PROTAC) molecule was designed for utilizing the ubiquitin–proteasome system (UPS) to degrade specific proteins by its bound E3 ubiquitin ligase. The model was proposed by Sakamoto et al. in 2001 [106], and subsequently rapidly and widely deployed for developing multiple new drugs, including anti-AR agents. ARV-110 (also known as bavdegalutamide) is one of them, which has exhibited a satisfactory effect in suppressing tumor growth and overcoming anti-AR drug resistance [107]. An ongoing phase I/II clinical trial has proved the safety of ARV-110 (NCT03888612) and its effect on cancer treatment (NCT05177042). More AR PROTAC degraders such as ARD-2585, ARD-61, ARCC-4 and A031 have also demonstrated anti-tumor potentials in pre-clinical studies [108,109,110,111].

Targeting AR by N-terminal inhibitors is another possible strategy. Some constitutively activated AR-Vs may lack the LBD but retain an intact NTD. Inhibitors targeting the NTD of the AR may inhibit not only the function of the entire length of the AR, but also the function of AR-Vs. One of the AR NTD inhibitors are the EPI compounds. In vitro and in vivo studies demonstrated that these compounds could suppress the growth of different prostate cancer models [83,87,88]. Clinical trials (NCT05075577, NCT04421222) have been proposed for metastatic or castration-resistant prostate cancer patients. Nuclear translocation inhibitors to prevent the AR from genomic transcription is another strategy to inhibit the function of the AR. As a transcriptional factor, nuclear translocation is the prerequisite for the action of the AR. Small molecules such as EPPI, CPPI and JJ-450 can suppress the nuclear translocation of the AR to block its transcriptional activity. These drugs have been evaluated to be effective in suppressing the proliferation of prostate cancer cells [86,112], but the effect on breast cancer remains unaddressed. Testing the effect of these chemicals on breast cancer may bring new insights into breast cancer treatment.

## 6. Conclusions

The roles of the AR should be clarified in different subtypes of breast cancer. Modulating the activity of the AR will be a promising strategy for treating breast cancer. However, more studies are needed to address the existing issues in the hope of translating these strategies into the clinical setting.

## Figures and Tables

**Figure 1 ijms-23-15342-f001:**
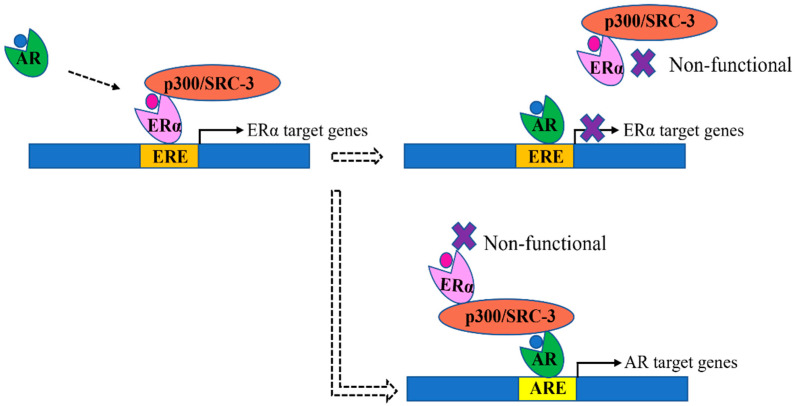
The proposed mechanism for the suppression of AR on ERα as reported by Hickey et al. [19]. Under normal circumstances, active ERα interacts with its coactivators, such as p300/SRC-3, and then bind to estrogen response element (ERE) to induce the transcription. Active AR may act as an ERα suppressor by competitive binding to ERE or by interacting with ERα coactivators to hijack ERα from ERE to ARE, thereby preventing ERα from transcribing its target genes.

## Data Availability

The materials and resources in this study are available from the corresponding author upon reasonable request.

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
