# Peer review of "Modulating the Activity of Androgen Receptor for Treating Breast Cancer"

_ijms, 2022, doi:10.3390/ijms232315342_

Round 1

Reviewer 1 Report

This is a comprehensive review on androgen receptor (AR) in breast cancer. Modulation of AR activity is proposed as a potential strategy for treating breast cancer. The Authors describe the different implications of AR positivity according to ER status, and accordingly I suggest they should report the following reference: 

Basile D, et al. Androgen receptor in estrogen receptor positive breast cancer: Beyond expression. Cancer Treat Rev. 2017 Dec;61:15-22. doi: 10.1016/j.ctrv.2017.09.006. Epub 2017 Oct 6. PMID: 29078133.

Author Response

Comments and Suggestions for Authors

This is a comprehensive review on androgen receptor (AR) in breast cancer. Modulation of AR activity is proposed as a potential strategy for treating breast cancer. The Authors describe the different implications of AR positivity according to ER status, and accordingly I suggest they should report the following reference:

 Basile D, et al. Androgen receptor in estrogen receptor positive breast cancer: Beyond expression. Cancer Treat Rev. 2017 Dec;61:15-22. doi: 10.1016/j.ctrv.2017.09.006. Epub 2017 Oct 6. PMID: 29078133.

Response:

Thank you very much for your appreciation and suggestion. This is a good reference that discusses the role of AR in ER-positive breast cancer. We have included it in our manuscript to enrich the content.

The related content was added in: “lines 75-78”.

Reviewer 2 Report

The manuscript is well-written and provides a comprehensive review of the potential mechanisms and applications of androgen receptor modulation for the treatment of breast cancer.

Specific comments:

1. The authors need to identify the nature of the review, i.e. narrative/systematic/scoping and provide a description of the strategies followed for literature search and inclusion.

2. The authors indicate that immunohistochemistry with AR may be informative for determining treatment strategies for breast cancer. In view of this suggestion, the authors need to elaborate on the methods for scoring AR immunohistochemistry and potentially tabulate the existing literature on the use of AR immunohistochemistry as a predictive marker for breast cancer,

Author Response

Comments and Suggestions for Authors

The manuscript is well-written and provides a comprehensive review of the potential mechanisms and applications of androgen receptor modulation for the treatment of breast cancer.

Specific comments:

Q1. The authors need to identify the nature of the review, i.e. narrative/systematic/scoping and provide a description of the strategies followed for literature search and inclusion.

Response:

Thank you for the suggestion. Our paper is a narrative review. To help the reader easier to identify the nature of this review, we have added a description of the strategies followed for literature search and inclusion in the following lines: “lines 54-57”

Q2. The authors indicate that immunohistochemistry with AR may be informative for determining treatment strategies for breast cancer. In view of this suggestion, the authors need to elaborate on the methods for scoring AR immunohistochemistry and potentially tabulate the existing literature on the use of AR immunohistochemistry as a predictive marker for breast cancer,

Response:

Thank you for the valuable suggestion. As the definition of AR positivity differed among studies, this might be due to the choice of antibody and scoring methods. We have included a table showing the antibody used and the corresponding scoring method. It will help the readers have a better understanding. Thus, we have suggested a standardized procedure which might avoid contradictory results. The table was added in: “line 279-280”
